# "We have to amplify what we saw at EBOVAC" – Assessing participant perceptions, attitudes, and acceptability of an ancillary care policy in an Ebola vaccine trial in the Democratic Republic of the Congo: A mixed methods study

Gwen Lemey [1,2]*, Ynke Larivière[1,2], Solange Milolo[3], Trésor Zola Matuvanga[1,3], Maha Salloum[1,2], Patrick Mitashi[3], Pierre Van Damme[2], Raffaella Ravinetto[4,5], Jean-Pierre Van geertruyden[1], Hypolite Muhindo-Mavoko[3], Vivi Maketa[3‡], Sibyl Anthierens[6‡]

1 Global Health Institute, Department of Family Medicine and Population Health, University of Antwerp, Wilrijk, Belgium, 2 Centre for the Evaluation of Vaccination, Vaccine and Infectious Disease Institute, University of Antwerp, Wilrijk, Belgium, 3 Tropical Medicine Department, University of Kinshasa, Kinshasa, Democratic Republic of The Congo, 4 Department of Public Health, Institute of Tropical Medicine Antwerp, Antwerp, Belgium, 5 School of Public Health, University of the Western Cape, Cape Town, South Africa, 6 Centre for General Practice, Department of Family Medicine and Population Health, University, of Antwerp, Wilrijk, Belgium

‡ These authors are joint senior authors on this work.
* Gwen.Lemey@uantwerpen.be

## Abstract

### Introduction

In a vaccine trial conducted between 2019 and 2022 in Boende, a remote, resource-constrained area of the Democratic Republic of the Congo, our research team developed an ancillary care (AC) policy to provide adequate care and follow-up for concomitant adverse events (AE), whether study-related or not. The trial aimed to assess the safety and immunogenicity of an Ebola vaccine regimen among approximately 700 healthcare providers and frontliners to strengthen outbreak preparedness in this Ebola-endemic region, where access to healthcare is severely limited by poverty, weak infrastructure, and an overstretched health system.

### Methods

A mixed-methods approach was used to assess participants' acceptability of the AC policy. First, participants with AE completed a questionnaire (1-—5 scale; 6 questions on AC policy support, 4 on the consequences of no support, and an open comment field). Second, a telephone survey (1-—3 scale; 3 questions evaluating the AC policy, 1 on unsupported AE and an open comment field) was conducted with participants, both with and without AE. Descriptive statistics were used for quantitative data

**Data availability statement:** The paper includes selected excerpts from open comment fields in the surveys and relevant interview transcripts. However, the complete data—such as audio recordings, full transcripts, and field notes—cannot be publicly shared due to the potential risk of identifying participants or revealing sensitive information. This data was not approved for public dissemination and was not included in the original study protocol. The data underlying the study's results is available upon reasonable request and approval by the relevant Ethics Committee. All data is securely stored on a long-term server maintained by the University of Antwerp and will be retained for 20 years to support future research or inquiries. For external access requests, please contact the university's data access committee at: RDM-support@uantwerpen.be.

**Funding:** The EBOVAC3 project has received funding from the Innovative Medicines Initiative 2 (IMI2) Joint Undertaking under grant agreement 800176. This joint undertaking receives support from the European Union's Horizon 2020 research and innovation program, the European Federation of Pharmaceutical Industries and Associations, and the Coalition for Epidemic Preparedness Innovations.

**Competing interests:** The authors have declared that no competing interests exist.

analysis, while open comments were coded qualitatively. Third, semi-structured interviews were conducted with participants who experienced a (serious) AE and either benefited from or did not benefit from the policy. Participants were selected using purposive and convenience sampling, and thematic analysis was performed.

## Results

Of 185 individuals with AE, 290 surveys were collected, with 93.5% expressing (very) strong appreciation for the AC policy. In the telephone survey, all 311 respondents supported the AC policy and emphasized its importance, 88.1% indicated it addressed their medical needs, and 35.7% reported experiencing an AE not covered by the policy. The 17 interviews revealed three major themes: 1) Experiences with AE management and AC support; 2) Financial impact of (non-) support; 3) Expectations of AC support. Participants who received AC reported personal, medical, and financial benefits, but noted limitations, such as the scope and duration of support, variations in local healthcare practices, and administrative hurdles.

## Conclusion

Both quantitative and qualitative findings show high endorsement for the AC policy support, regardless of participants' personal use. This acceptability study highlights the importance of AC in clinical trials and comprehensive participant care in research.

## Introduction

Between December 2019 and October 2022, the Universities of Antwerp (UAntwerp) and of Kinshasa (UNIKIN) carried out a phase 2 vaccine trial, referred to as 'EBOVAC3', to assess the safety and immunogenicity of an Ebola vaccine regimen administered to 698 registered healthcare providers and frontliners [1,2]. It aimed to improve the Ebola-outbreak preparedness of the Boende region, a remote, resource-constrained and Ebola-endemic area in the Democratic Republic of the Congo (DRC). The area is characterized by its rainforest and poor, often impassable road infrastructure. Overall, the DRC health system is under considerable strain due inadequate resources, insufficient qualified healthcare personnel, poor infrastructure, and a high disease burden [3]. An estimated 79% of the population lives below the national poverty line [4]. The government's universal health coverage initiative, implemented with bilateral partners, focuses primarily on mothers, newborns and children [5]. While decentralizing health services is a policy goal, access to healthcare remains challenging for much of the population and largely depends on patients' out-of-pocket payments [6], which strain household incomes and deepen poverty.

To safeguard the rights, safety and well-being of trial participants, guidelines and directives outline the responsibilities of the different research stakeholders, e.g., ethics committees, investigators, and sponsors. For instance, the Council for International Organizations of Medical Sciences recommends the provision of ancillary care (AC) as

a way to balance the burdens and benefits of research [7]. However, such guidelines provide a conceptual ethical framework rather than practical guidance for implementation. The application of the ethical values in practice, taking into account different research settings and local variables, needs to be contextualized, and continues to challenge sponsors and investigators [8].

Due to the limited quality and affordability of health care services available in the Boende region, the EBOVAC3 sponsor and principal investigator provided AC to address, treat and support trial participants experiencing medical events, irrespective of a causal link with the research intervention [9,10]. A study-specific AC policy was designed to medically and financially cover participants' medical conditions, but context-specific challenges were met [11,12]. In November 2021, before the start of the trial's last operational year (3 years in total), the policy was introduced in the trial. Prior to the implementation of the policy, (serious) adverse events ((S)AE) were handled on a case-by-case basis. Following AC policy implementation, all newly reported (S)AE, both related and unrelated to the investigational product or the trial, were systematically managed in accordance with the policy and associated algorithm(3). SAEs that occurred before the policy was introduced could still be 'retrospectively' supported, provided proof of payment was available. Possible support outcomes included; 1) the provision of medication and/ or diagnostic tests from the study pharmacy; 2) direct payment of medical invoices or of medication from external pharmacies; 3) reimbursement of prefinanced medical invoices; 4) a combination of support outcomes, or 5) no support.

The evaluation of the AC policy followed a mixed methods approach. The first objective was to assess the policy's implementation, utilization, and feasibility, while the second objective focused on exploring participants' perceptions, attitudes and acceptability of the policy. In previous publications, we reported on participants' use of the AC policy, including medical and financial support outcomes, geographical determinants, budgetary implications, as well as on implementation challenges [12,13]. However, a critical gap remained in understanding how participants experienced and perceived the AC policy itself. To address this, we conducted an in-depth exploration of participants' views and acceptability of the policy, thereby complementing our earlier quantitative analysis and offering a more comprehensive understanding of the AC policy's reception and perceived impact among trial participants.

## Methods

### Study setting

The Ebola vaccine trial, as well as the acceptability study of the trial's AC policy, were conducted at the general reference hospital (GRH) of Boende. Trial participants were healthcare providers and frontliners from different health districts of the Tshuapa province. The former are considered educated healthcare professionals in the formal healthcare sector, while frontliners, such as community health workers and Red Cross volunteers, provide direct patient or community support and emergency care, often having acquired basic training and frequently working without pay from health authorities [14].

### Study design, sampling, and data collection

This was a longitudinal, mixed-method study with a recurrent cross-sectional design, embedded in an open-label, single-centre, randomized Ebola vaccine trial (EBOVAC3; clinicaltrials.gov: NCT04186000). The vaccine trial was set up to evaluate the safety and immunogenicity of the Ad26.ZEBOV, MVA-BN-Filo vaccine regimen and of two different booster vaccination arms [2]. The AC policy and its acceptability study were implemented from November 2021 until October 2022, coinciding with the final year of the 3-year vaccine trial, and included a total of 655 participants that were still enrolled in the vaccine trial at that time. Both quantitative and qualitative data collection were conducted at pre-established regular intervals. Fig 1 illustrates the timeline of scheduled trial visits per trial arm and data collection timepoints of the AC acceptability study.

### Surveys for participants who experienced (S)AE

The surveys focused on trial participants who had experienced AE or SAE, and had either benefitted from the AC policy or had not. There was no target sample size; all participants with (S)AE were invited to take the survey and were included.

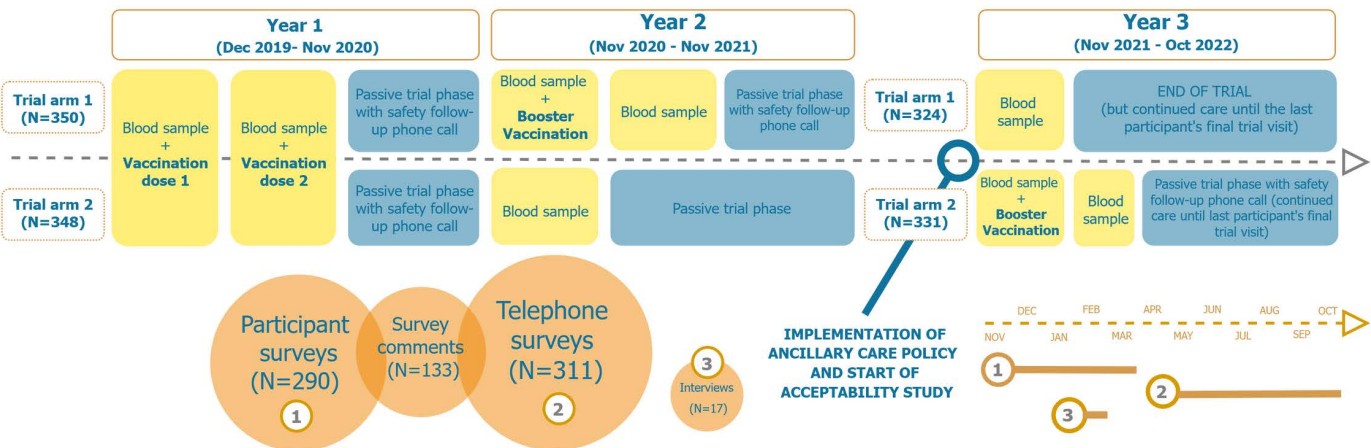

**Fig 1. Study design of the Ebola vaccine trial, including data sources and collection points of the ancillary care acceptability study.** Trial arm 1 and Trial arm 2 indicate different trajectories within the trial. Yellow boxes represent scheduled visits during active trial phases; while blue boxes represent passive trial phases. Orange circles denote the three data collection sources in the acceptability study, with their sequence and duration represented on a dotted bar.

Data collection took place between November 2021 and March 2022, when trial participants came to the GRH for either treatment or reporting of their previously treated medical condition. Depending on the participant's preference, the survey form was either in French or Lingala. Survey forms were completed by members of staff. Surveys consisted of 10 questions (grade 1–5) with the option to provide open comments. Participants were invited to respond to 6 questions about their experiences with and appreciation of the AC policy support, and an additional 4 questions on their treatment options and potential consequences aside from the trial's AC support. The surveys were drafted by GL (woman, Belgian, PhD student UAntwerp) and AP (woman, Italian, post-doctoral social scientist UAntwerp) performed quality control on the drafted survey form. The questions were translated (French to Lingala) and back-translated (Lingala to French) by TZM (man, Congolese, PhD student UNIKIN) and SM (woman, Congolese, UNIKIN staff).

## Interviews

The interviews focused on participants who had experienced a(n) (S)AE, and had either benefitted from the AC policy or had not. Data collection took place during a field visit of UAntwerp researchers to Boende in February 2022. The interviews were conducted by GL at the GRH, with the assistance of a study nurse as translator. The interview guide, developed based on the survey questions and tailored for live, one-on-one conversations, was used to explore these questions in greater depth during the interviews. Potential candidates for interviews were contacted by telephone by the clinical trial staff.

Participants were selected using purposive and convenience sampling. First, the study data manager performed a randomized, unbiased selection from the database reporting participants' medical events. The selection criteria were that participants had one or more (S)AE and were still actively enrolled in the trial (e.g., not deceased, not lost to follow up or withdrawn consent). Secondly, purposive sampling was used to enhance case diversity. Selection criteria included participants' residential locations, treatment venues, types of treatment received (e.g., conventional medical care or traditional medicine), and the nature of support outcomes (such as reimbursement, direct payment, provision of pharmacy medication, or no support). Thirdly, convenience sampling was used, based on recommendations from the study staff. This approach facilitated the recruitment of participants who were present at the study site during the interview period, thereby optimizing logistics and enabling the inclusion of a broader range of perspectives.

The interviews were audio recorded with participants' consent. The questions were initially posed in French, with a study nurse (RAM or BLB) serving as an interpreter, translating into Lingala. The latter were female staff members native to the Tshuapa province, providing linguistic and cultural contextuality to the interview process. Throughout the interviews, the researcher (GL) maintained a reflective journal for field notes. These notes captured methodological observations, including the possibility of suggestive or biased translation of research questions and answers, along with reflexive considerations on how both the interpreter and the researcher might influence participants' responses.

The assessment of data sufficiency was conducted within the constraints of the field research context. Pragmatic considerations, including the limited duration of the field visit, participants' accessibility to the study site, and their work commitments, influenced the data collection process. Given these constraints, a conventional iterative process of data collection and analysis was not feasible. Instead, GL employed a pragmatic approach, conducting a concentrated period of interviews during the field visit. While this approach diverges from the ideal iterative process, it reflects the realities of field research in challenging settings. Data sufficiency was evaluated based on the richness and depth of information obtained from the interviews, the emergence of recurring themes, and the extent to which the research questions were addressed. This assessment was made at the end of the field visit, acknowledging both the limitations and the practical necessities of the research context [15,16].

## Telephone surveys

Telephone surveys facilitated not only the inclusion of participants with reported (S)AE, but also of participants who had *not* experienced a(n) (S)AE during active policy implementation. During the trial's last safety follow up telephone contact, conducted between May and October 2022, participants were invited to respond to 3 questions (scale 1–3) related to the AC policy support. However, only participants from trial arm 2 could be included (target sample size = 331), as the arm 1 participants already had their last safety phone call at a different timepoint (Fig 1).

The surveys were short and simple as not to overburden the workload of the healthcare staff and taking into account the telephone network challenges in this remote setting. The questions enquired about 1) participants' endorsement of the AC policy, 2) their assessment of the policy's response to medical needs, and 3) the perceived importance of AC. The questions were drafted by GL and (back-)translated (French to Lingala to French) by TZ and SM. The telephone contacts were done by EE (male, Congolese, local trial staff); questions were posed in either French or Lingala, depending on the preference of the participant.

## Analysis

The qualitative analysis was led by GL under the supervision of SA, employing a rigorous thematic analysis approach [17,18]. The verbal narratives from the interviews, which were conducted in a combination of French and Lingala, were transcribed verbatim by a female physician based in Boende. This transcriptionist also translated the Lingala questions into French, ensuring linguistic consistency across the dataset. Data familiarization was achieved through multiple methods. GL immersed herself in the data by listening to the audio recordings, which allowed for a nuanced understanding of tonal and contextual elements. This was complemented by repeated readings of the French transcripts, survey comments, and field notes, providing a comprehensive overview of the data.

The thematic coding process was inductive. Relevant data pertaining to the research question were systematically coded. This process was iterative, involving initial coding across a subset of interviews, followed by the grouping of codes into potential themes. Theme development involved reviewing the created themes in relation to the coded extracts and the entire dataset. This process resulted in the creation of a thematic 'map' of the analysis. Subsequently, themes were refined, with clear definitions and names generated for each.

Throughout the analysis, flexibility and reflexivity were maintained, ensuring that the focus remained on pragmatically addressing the research question. The qualitative data analysis software NVivo (Release 14.23.3) was utilized to support the coding and theme development processes, enhancing the systematic nature of the analysis. YL conducted the quantitative analysis using R. Descriptive statistics were used to present the data in number (%), mean (SD), or median (range). All statistical analyses were performed in R (version 4.3.1). GL used these descriptive results to create figures presented in this manuscript using Excel and Draw.io.

### Ethical considerations

Ethical approval was obtained from the National Health Ethics Committee in DRC for the AC policy (n°231/CNES/BN/PMMF/2021) and the AC evaluation study (n°313/CNES/BN/PMMF/2020). Informed consent was obtained from all individuals who participated in the acceptability study, including oral informed consent for audio recording.

### Results

Table 1 presents the demographics and baseline characteristics of research participants who took part in the acceptability study via the telephone surveys, the surveys after experiencing (S)AE, and the interviews. We previously published findings on the association between demographic and baseline characteristics and (S)AE reporting—namely, the lack of influence of age and medical history at the start of the study, lower reporting rates among men and health facility workers, and higher reporting in Arm 2 likely due to more scheduled visits [13]. The following paragraphs present the findings from each data collection method. First, the results from both surveys are presented, followed by the interviews, which are organized into thematic summaries supported by quotes.

### Survey results

For the telephone surveys, 311 Arm 2 participants (Fig 1) – both with and without (S)AE – responded to three short questions (scale 1–3), with the option to provide comments (N = 58). All respondents (100.0%) indicated to support the (medical and/or financial) AC provisions for AE and also indicated that these were important to them. A total of 274 participants (88.1%) felt that the AC provisions met their medical needs during trial participation, regardless of whether they had used the support or not. A total of 111 participants (35.7%) indicated to have endured an AE that was not supported by the study's AC policy.

From participants who had experienced (S)AE, a total of 290 surveys were collected among 185 individuals. Fig 2 illustrates the scores given to questions 1–6. The overall appreciation of the AC support was high (Q1); 93.4% (n = 271) indicated moderate to high satisfaction, whereas 89.9% (n = 261) of the participants evaluated their expectations on AE management being (fully) met (Q2), and their health needs (entirely) fulfilled (Q3). Additionally, 72.4% (n = 210) expressed their well-being as moderate to high during research participation (Q4). When enquiring about the added value of AC provisions (Q5), 93.0% (n = 270) marked them important to very important during research participation. Lastly, the AC provisions were viewed to improve participants' health status (Q6); 91.3% (n = 265) indicated a positive to very positive contribution. A total of 75 participants provided open comments in the survey form.

The remaining questions (7–10, S1 Fig) enquired about participants' health-seeking behaviour in the absence of AC policy support. A total of 96.9% (n = 281) and 95.9% (n = 278) respondents expressed that they would have still sought and found treatment for their AE, respectively. When asked about the possible financial consequences of non-support, 74.2% anticipated an impact ranging from minor (61.4%; n = 178) to major (12.8%; n = 37). In the absence of AC support by the study, most respondents (93.5%, n = 271) indicated their preferred place of treatment of their medical event to be the GRH in Boende.

### Interview results

To conduct the interviews, we initially selected 16 participants through purposive sampling: 8 participants with AE and 8 with SAE. To further enhance diversity, we selected an additional 6 participants, consisting of 2 with AE and 4 with SAE. In total, 22 individuals were invited for one-on-one interviews, and 12 agreed to participate. During the interviews, we

**Table 1. Demographics and baseline characteristics of trial participants in the acceptability study (n = 655).**

| | Participants *without* reported AE (N = 285) | Participants who reported ≥ 1 AE (N = 370) | Surveyed participants with AE (N = 185) | Telephone survey participants *without* self-reported AE (N = 200) | Telephone survey participants *with* self-reported unsupported AE (N = 111) | Interviewed participants with (S)AE (N = 17) |
|---|---|---|---|---|---|---|
| **Sex, n (%)[a]** | | | | | | |
| Female | 57 (20.0) | 90 (24.3) | 53 (28.7) | 166 (83.0) | 21 (18.9) | 4 (23.5) |
| Male | 228 (80.0) | 280 (75.7) | 132 (71.4) | 34 (17.0) | 90 (81.1) | 13 (76.5) |
| **Age** | | | | | | |
| Median | 45.0 (20.0-75.0) | 47.0 | 47 | 45.5 | 47.0 | 46 |
| Mean | 44.6 (11.6) | 45.5 | 46.6 | 44.8 | 46.1 | 48.2 |
| **Profession all categories, n (%)** | | | | | | |
| Community health worker | 103 (36.1) | 122 (33.0) | 44 (23.8) | 71 (35.5) | 37 (33.3) | 1 (5.9) |
| Nurse | 87 (30.5) | 83 (22.4) | 46 (24.9) | 48 (24.0) | 36 (32.4) | 9 (52.9) |
| First aid worker | 52 (18.3) | 109 (29.5) | 63 (34.1) | 52 (26.0) | 22 (19.8) | 5 (29.4) |
| Hygienist | 13 (4.6) | 23 (6.2) | 13 (7.0) | 11 (5.5) | 3 (2.7) | 0 (0.0) |
| Midwife | 10 (3.5) | 18 (4.9) | 10 (5.4) | 5 (2.5) | 4 (3.6) | 1 (5.9) |
| Doctor | 6 (2.1) | 5 (1.4) | 2 (1.1) | 6 (3.0) | 1 (0.9) | 0 (0.0) |
| Health facility cleaner | 6 (2.1) | 4 (1.1) | 4 (2.2) | 4 (2.0) | 3 (2.7) | 1 (5.9) |
| Care giver | 5 (1.8) | 2 (0.5) | 2 (1.1) | 2 (1.0) | 3 (2.7) | 0 (0.0) |
| Lab technician | 2 (0.7) | 0 (0.0) | 1 (0.5) | 0 (0.0) | 0 (0.0) | 0 (0.0) |
| Pharmacist aid | 0 (0.0) | 2 (0.5) | 0 (0.0) | 0 (0.0) | 1 (0.9) | 0 (0.0) |
| Other | 1 (0.4) | 2 (0.5) | 0 (0.0) | 0 (0.0) | 1 (0.9) | 0 (0.0) |
| **Medical history, n (%)** | | | | | | |
| Yes | 52 (18.3) | 73 (19.7) | 41 (22.2) | 18 (9.0) | 96 (86.5) | 1 (5.9) |
| No | 233 (81.8) | 297 (80.3) | 144 (78.0) | 78 (39.0) | 15 (13.5) | 16 (94.1) |
| Unknown | 0 (0.0) | 0 (0.0) | 0 (0.0) | 104 (52.0) | 0 (0.0) | 0 (0.0) |
| **Trial arm, n (%)** | | | | | | |
| Arm 1 | 169 (59.3) | 156 (41.6) | 87 (47.0) | 0 (0.0) | 0 (0.0) | 8 (47.1) |
| Arm 2 | 116 (40.7) | 216 (58.4) | 98 (53.0) | 200 (100.0) | 111 (100.0) | 9 (52.9) |

(S)AE = (serious) adverse event; N = the total number of participants in a given category; n (%) = the number (percentage) of the participants corresponding to the demographic or baseline characteristic category.

[a]The higher absolute number of AE reporting in males reflects the higher proportion of males in the Ebola vaccine trial at the time of the acceptability study (n = 508; 77.6% male versus n = 147; 22.4% female) [13].

recruited 5 more participants as they were on-site. Altogether, 17 participants took part in the study. This diverse sample ensured a wide variety of perspectives and experiences with the AC policy. The interviews lasted between 15 and 30 minutes.

Overall, the comments provided in the two surveys, as well as the answers given during the interviews, could largely be grouped into three broad themes: 1) Experiences with AE management and AC support; 2) Financial impact of (non-) support; 3) Expectations of AC support.

## 1. Experiences with AE management and AC policy support

### 1.1 Appreciation and support for provided care

Participants expressed strong appreciation for the AC policy, as evidenced by both survey responses and interviews with those who experienced medical events. The high level of satisfaction was frequently mentioned in the open comment

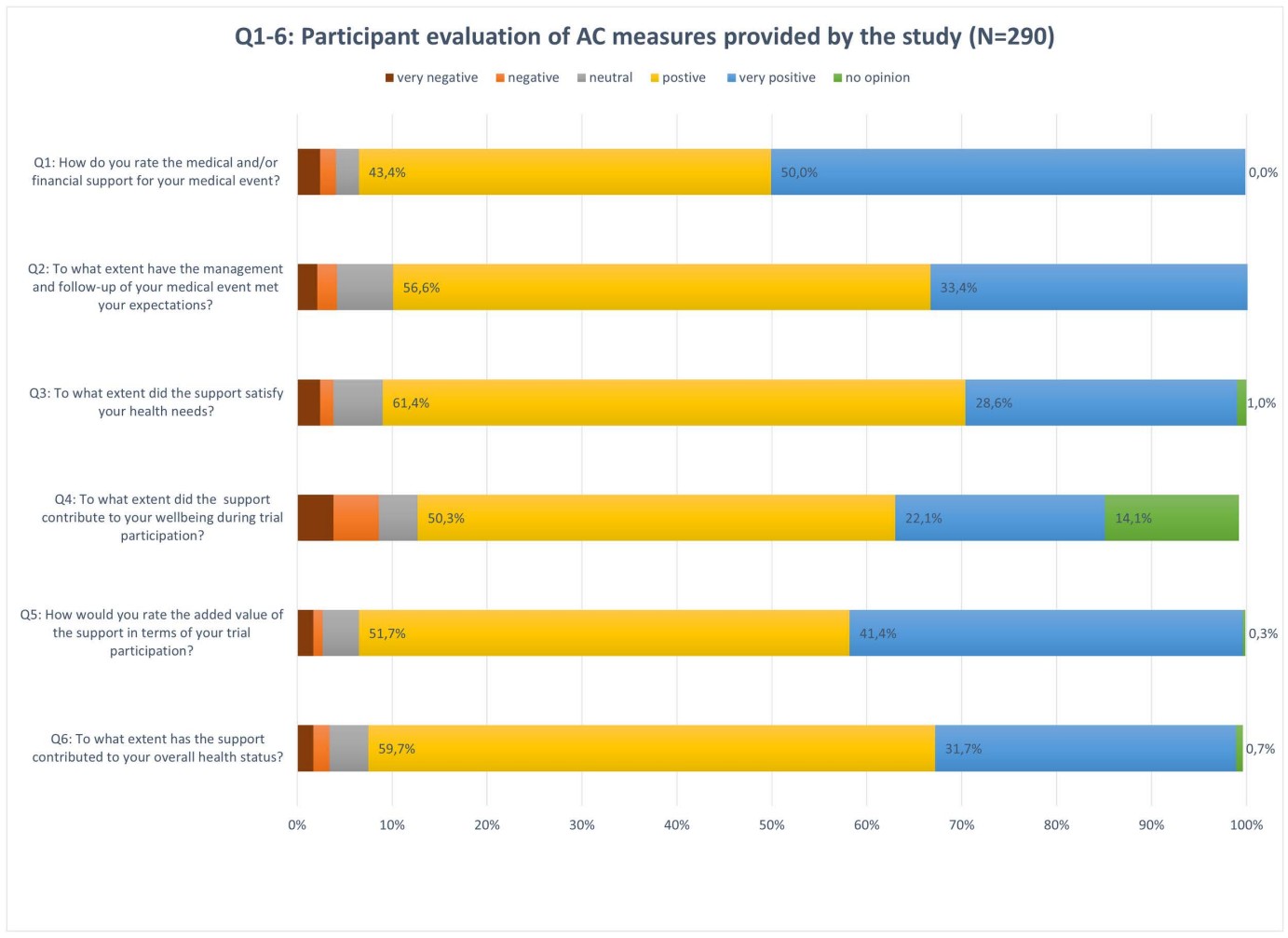

**Fig 2. Survey ratings (questions 1 to 6, scale 1-5) of participants with adverse events (AE) evaluating an ancillary care (AC) policy.**

sections of the surveys and echoed during interviews. Participants consistently valued the support provided by the study, particularly in terms of health care coverage and assistance with medical expenses:

*"We support the idea of taking charge of our health."* (Survey respondent, participant with medical event, male)

*"I'd like to thank [the study] for helping me pay for hospitalization and for the medical assistance with medication."* (Survey respondent, participant with medical event, male)

The appreciation extended beyond mere gratitude for the assistance; participants emphasized the reliability and responsiveness of the care provided. They felt reassured by the study's commitment to their well-being, with some suggesting that expanding these efforts could further enhance the positive impact:

*"It was good. They have given us medicines, no matter at what time we got sick. We would call and they would receive us without any problems. And if they add to that, it will be even better.* (Interviewed participant with SAE, male)

Participants also recognized the value of the AC policy as a model of care that could be built upon in future studies:

*"Well, in relation to that, what EBOVAC has done is a good thing, and if another person comes along, we have to amplify what we saw at EBOVAC."* (Interviewed participant with AE, male)

1.2  Perceptions of impact on health and well-being

The majority of participants reported positive experiences, especially in terms of the immediate care and follow-up provided during their medical event by the study. They also highlighted the perceived direct health benefits resulting from the AC policy:

*"Your follow-up was good, I mean at least 10/10, you followed us. So 10/10, because if I was ill, I was looked after by you, in any case I was fine. In the event of illness, as soon as I make the trip, leave Boende for another place, as soon as I return, they reimburse the money of treatment when I was ill, in any case that's normal."* (Interviewed participant with AE, male)

*"I was happy when you paid us money, but especially when you took care of us. When I got sick, I would come in, I would be taken care of, and you would monitor our health."* (Interviewed participant with SAE, female)

However, some participants emphasized the perceived limitations of the support, particularly the short-term nature of the health benefits, noting the contrast between the immediate relief obtained and the long-term concern about their health and well-being during research participation.

*"Whatever it is, this help doesn't help me every day. The little we were given, we use today, and tomorrow it's over. Today, and tomorrow it's over. But to say that it helped me for a while, many months apart? Never."* (Interviewed participant with AE, male)

1.3  Timing and Implementation challenges of AC policy

The delayed introduction of the AC policy during the final operational year of the Ebola vaccine trial presented significant challenges that impacted participants' experiences. Many participants had already encountered unrelated (S)AE prior to the policy's implementation, which affected the management of their medical conditions and shaped their overall perception of the support provided by the study. One participant expressed frustration that the AC policy was not in place earlier, highlighting the gap in care during the early stages of the trial:

*"They should have started earlier. In those intervals, you can get sick too. But the last ones they took charge of us, they guaranteed only a six-month term. But from the beginning, when we'd started, we were getting sick as well."* (Interviewed participant with AE, male)

Another participant recounted a particularly distressing experience where the delayed implementation of the AC policy could not prevent a long-term disability. This account underscores the profound impact that the timing of AC policy implementation can have on participants' lives:

*"No, it didn't add anything, it rather almost killed me. [...] Because we should have decided from here [...] to arrange [my problem]in Kinshasa. [...] But, as it took, it's now been, two years? A year? [....] But if it was in the beginning, it would have been fixed. That's how I ended up disabled!"* (Interviewed participant with SAE, male)

In addition to the timing issues, participants frequently cited concerns about the unavailability of necessary medications and diagnostic tests, as well as the perceived inadequacy of the medical support provided. These gaps in care further eroded trust in the AC policy and the study itself:

> *"You promised us that when someone gets sick, 'We will support it'. But in the course of the trial operations, many of the personnel [study participants] fell ill, and there was no medicine here. […] We came, no medicine. But when you make agreements, you have to respect them, and you haven't respected that!"* (Interviewed participant with SAE, male)

> *"For this problem, I don't know the content of the products that were in the pharmacy, but what I'm saying is the doses, the dose I was given is insignificant."* (Interviewed participant with AE, male)

> *"Lack of in-depth tests. Add more tests."* (Telephone survey respondent, female)

1.4  Interaction with Local Healthcare Context and Practices

When developing the study's AC policy, we tried to take into account the local practices, including the fact that the family generally provides for patient's non-medical needs. However, participants expressed a desire for nutritional support. Some highlighted the fact that food was not provided as part of AC during hospitalization, indicating that their expectations were not fully met:

> *"On the medication side, it was fine. But if someone was hospitalized, they weren't given any food…"* (Survey respondent, participant with medical event, female)

The AC policy was designed to align with the local standard of care available in the research setting, which also shaped how participants' AE were managed and followed up. This alignment sometimes led to gaps in care, particularly when local healthcare facilities lacked the capacity to address more complex medical needs:

> *"Yes, you can see that my illness required a lot of money. And that's because I didn't know they were doing it here. Because they said that the general hospital wasn't able to do it."* (Interviewed participant with SAE, female)

## 2. **Financial impact of (non-)support**

2.1 Modalities for reimbursement

Participants generally expressed appreciation for the financial support for medical expenses incurred outside the study hospital, or prior to the policy's official start. However, a significant point of concern revolved around the requirement of providing proof of payment, as a pre-condition for reimbursement. Some participants reported that their inability to provide such proof resulted in incomplete reimbursement, leading to financial strain and debt, as illustrated by one participant stating they were 'not happy' due to the 'little' support received despite significant expenses. Another participant suggested the need for 'full coverage' during research participation. Overall, the financial support was generally positively valued, but the non-negotiable requirement for proof of payment led to dissatisfaction for some.

> *"I wasn't happy about that, because the help was only trickling in. I wasn't happy about that. The lady here had come all the way down to the house […] and I told her that I wasn't happy about it because I'd gone into a lot of debt. […] Even though I had paid a lot of money for the operation. Well, with the little you're giving me, I really didn't like it."* (Interviewed participant with SAE, male)

*"You did not take full charge of my hospitalization. Suggestion: one must be fully covered during your studies."* (Survey respondent, participant with medical event, female)

## 2.2 Importance of transportation costs

At times considerable distances had to be traveled by participants to reach the study site. Some criticized the modalities of transport compensation as an issue, foreseen only for scheduled visits. As such, participants who came to the trial site for AE reporting or management during an unscheduled visit did not receive compensation for their transport costs:

*"Since we're here, I'd like to propose the following. They took care of us, but at one point we were sick all the time and we could die like that. Because there wasn't really any money for transport. Even at night. You should either give transport, or come and take us or whatever."* (Interviewed participant with SAE, female)

*"Next time, pay for transport when we come to withdraw the medicines, even if you had not invited us."* (Survey respondent, participant with medical event, female)

Conversely, on several occasions it became clear that the transport contribution for scheduled study visits had contributed to other aspects of participants' lives, pointing to its importance:

*"It's helped us, first because we know you, and second, we have had the opportunity to have free medicines. If I fall ill, I come and they give me the medicines for free. But also, transport, that's helped us a lot, for the little problems I had. When I come, they give me this $50, and that's really helped us."* (Interviewed participant with SAE, female)

## 2.3 Burdens of AE on personal means and debts taken

When enquiring about participants' personal means for AE treatment aside the study's AC policy, participants often expressed a determination to get the needed treatment, taking different strategies to accomplish that:

*"We would always have found a solution. I'm a man, I've got the livestock, but there are also household goods that we buy. Really, I would just set up a business so that I can be looked after. I wouldn't have any problems with that."* (Interviewed participant with SAE, male)

*"I would have had treatments, even ancestral ones, even medical ones. Yes, all of them!"* (Interviewed participant with SAE, male)

Moreover, the general study allowances, as well as financial support and reimbursements for prefinanced medical invoices, often facilitated the repayment of loans and debts participants had taken on for AE treatment:

*"Well with this money… I had also taken on debts. I had gone into debt to other people to get treatment. I paid it back!"* (Interviewed participant with SAE, male)

## 2.4 Impact on livelihoods

The reimbursements provided for medical expenses incurred at various stages and across different healthcare facilities had broader, often unanticipated, positive impacts on the lives of participants and their families. These financial reimbursements not only alleviated the burden of medical costs, but also extended support to other essential aspects of their daily lives, improving overall well-being.

*"It helps me sometimes to buy medicine, like I had bought. Maybe I borrowed some money, or my parents gave it to me. And it's a joy that I was in EBOVAC, look what I got for treatment!* (Interviewed participant with AE, female)

For some participants, the refunded medical costs provided a much-needed financial relief that transcended healthcare, enabling them to address other pressing needs. For instance, one participant highlighted how the reimbursement allowed her to manage everyday expenses, such as food and housing, thereby contributing to her financial stability:

*It will be a joy to the family. It's going to help me eat, but also where I rent, my accommodation. So a refund helps financially."* (Interviewed participant with SAE, female)

Another participant emphasized how the financial support eased the pressure of paying for school fees and personal expenses, underscoring the broader economic impact of these reimbursements:

*"Yes, because sometimes I'd run out of the child's money [school fees], or for myself. So that helped, and added. It's good."* (Interviewed participant with AE, male)

### 3. Expectations of AC support

#### 3.1 Medical support expectations

All participants enrolled at the time of policy implementation were informed of the AC policy during an informed consent procedure, stipulating the modalities, scope and duration of support. However, some surveyed and interviewed participants indicated that the AC support provided during the Ebola vaccine trial did not fully meet their expectations in terms of the extent of the provided medical support.

*"I expected… Let's say it wasn't enough. […] For example, I thought that since I was pregnant, sometimes [the study] would take care of everything. Whether it's childbirth or whatever, that's what I had in mind. But I spent a lot of money, a lot, because during the whole pregnancy, there were only problems.* (Interviewed participant with SAE, female)

A few participants, being healthcare providers themselves, offered ideas and suggestions for the study's AE management and AC support in future studies:

*"We've come a long way, taking care of medication and transport. In the near future, I'd like them to set up their pharmacy and take a look at our pharmacy in Tshuapa, for example. They know that such and such a case is common, and that we need to be able to provide such and such medicines. This will be for the care of the study participants."* (Interviewed participant with SAE, female)

Others had expectations that highlighted some limitations of the AC policy that were related to the standard of care available in this remote, resource-constrained setting:

*"I suggest that the treatment should continue for a while and that we should look into ways of treating illnesses that are not treated in Boende, such as eye disease."* (Survey respondent; male, participant with medical event)

*"Make a contract with our village structures for care; here it's far away."* (Telephone survey respondent without a reported AE, male)

## 3.2 Financial support expectations

As seen, some participants had hands-on experiences with the AC policy's limitations with regards to its financial support modalities. Naturally, some participants expressed desires and expectations that would better cover their financial needs in the framework of an (S)AE, which had remained unmet:

*"It wasn't what I expected. What I had expected and what I saw was different.[…] I expected that if you got sick, you'd come to the hospital, get some medicine and they'd hospitalize you and give you your meals. They take care of you. Nothing, nothing special, whether it's transport… We thought we were going to get a lot of money, nothing".* (Interviewed participant with SAE, female)

*"We're not really happy because little support; add more financial support next time."* (Survey respondent; participant with medical event, male)

## 3.3 Expectations beyond the modalities of the study and AC policy

Although data collection occurred during the final year of the Ebola vaccine trial, some participants expressed expectations and made requests that extended beyond the anticipated financial and medical support for their medical events. These expectations also exceeded the intended scope and duration of the trial, with hopes that the study would continue providing assistance in the (near) future:

*"I suggest that [the study] continues to help the community that suffers during a period of epidemic, for example, and that does not have the means to escape this epidemic or pandemic."* (Survey respondent; participant with medical event, male)

*"I would like to inform you not to abandon us and to continue to help us and even more as you have done other times."* (Survey respondent; participant with medical event, female)

Some participants voiced specific expectations tied to their personal circumstances, reflecting broader socio-economic vulnerabilities:

*"I don't have a job, I don't have a father or mother, I don't have a good husband. I have three children who go to a public school. In this school, we pay per month. That's why I'm asking you to help me with my finances, so that I can help my three children. And through the doctors, so that it stays that way, so that it can help my health, so that I have the strength to help these children."* (Interviewed participant with AE, female)

Others reflected on potential long-term effects of the vaccine or anticipated the need for financial support to manage future medical expenses after the study's completion:

*"You, you're going to leave. The study is over, it's over now, but in the end, the side effects that will come afterwards... We could have something, we can. How are we going to be able to pay for the drugs elsewhere? You see, drugs are expensive outside. It's expensive to buy them privately! Will we get a little present at the end? Then we will be able to buy medicines for the other effects that will come."* (Interviewed participant with AE, male)

*"[…] when it happens that they don't work here anymore, as the study will also come to an end. So, as the study goes, I can have reactions after the study. And if I came here and the doctors told me that it was already over, what would I do then as far as my own care was concerned?"* (Interviewed participant with SAE, female)

## Discussion

This study aimed to explore participants' perceptions and attitudes towards the AC policy implemented in the Ebola vaccine trial conducted in Boende, DRC. In recent years, AC policies have become a relevant focus of evaluation within global health research. Our research group has previously contributed to this discourse by publishing the results of a formal quantitative evaluation of our study-specific AC approach [13], as well as by documenting the development process and ethical challenges associated with its implementation during the Ebola vaccine trial in Boende [11,12]. Other researchers have similarly explored the landscape of AC through various qualitative and systematic reviews, including studies on stakeholders' perspectives in Malawi [19], benefit-sharing practices in Kenya [20], principal investigators' views on the ethics of medical care in clinical research [21], and AC practices in East and Southern Africa [22].

To the best of our knowledge, this is the first study to examine the acceptability of a study-specific AC policy from the perspective of the participants themselves, offering insights into their views and practical experiences as potential beneficiaries. A noteworthy finding is that participants did not consistently distinguish between specific AC provisions and the general care provided for AE management and follow-up within the study. For instance, medical support offered under the AC framework – intended as supplementary, non-research-related care – was often perceived by participants as an integral aspect of their involvement in the study. Conversely, financial contributions, such as reimbursements for pre-financed medical events or compensation for time and transport related to scheduled visits, were more readily recognized as part of the AC support. Furthermore, when participants explicitly referred to the study, this did not substantially influence their appreciation of AC provisions, despite the frequent linkage between the two.

Overall, there was a high participant acceptability of the AC policy within this trial. Similarly to what our research team, as well as other studies, have pointed out, the trial participants highly valued the access to healthcare at the research site when they were ill or experienced medical events [23–25]. It is also noteworthy to point out that the policy evaluation via telephone surveys involved both participants who had and had not experienced AE, contributing to participant validation of the overall findings; both subgroups were appreciative of the AC policy in place during study participation. The qualitative findings (survey comments and interviews) thus support the quantitative results arising from the surveys; participants expressed a high to very high endorsement of the trial-specific AC provisions.

Some challenges mentioned by participants referred to the practical organisation and planning of AC, such as the timing of policy implementation, pharmacy stock-outs and administrative, proof-of-payment requirements. Others were related to the characteristics of the local setting and existing healthcare infrastructure, on which the policy was tailored. The policy had overlooked certain cultural expectations and resource gaps, such as the provision of food during hospitalization, the lack of specialized care locally, and the travel costs for unscheduled visits. As such, including food provision in the AC policy would improve recovery and overall well-being, covering travel costs would enhance access and reduce financial strain, flexible verification methods for prefinanced medical invoices would ensure inclusivity, and incorporating support for accessing specialized care, such as referrals or financial assistance, would ensure participants receive the comprehensive care they need. Therefore, it is crucial to consider local healthcare practices and limitations when designing and implementing AC policies, as they impact healthcare accessibility and equity.

It is known that the opportunity for a free medical check-up before inclusion in the study, or the free medical care provided during the study, can be a motivating factor for some participants to enroll in the research [8,26,27]. Additionally, the narratives of our participants indicate that research participation may offer other benefits not accounted for by the trial itself, as these advantages exceed the medical aspects and are tied to a specific socio-economic setting [26]. For instance, it was pointed out that the reimbursement of (S)AE treatment costs, or other study-related financial contributions, led to the repayment of loans made for AE treatment, or positively impacted their livelihoods (e.g., paying school fees or rent).

Next to, amongst others, accessing AC services and contributing to science and health, financial benefits are often considered to be one of the primary motivations for trial participation by healthy volunteers. This is considered legitimate if the remuneration is fair, appropriate and subject to ethical review [24,28,29]. In our study, participants reported limited

financial means for AE care, and the financial impact of such care was considerable. Despite being active healthcare providers and frontliners, their wages were low – consistent with the general figures in the DRC, particularly among public servants. Acute medical conditions can, therefore, severely threaten household finances and negatively affect the health and well-being of many [6].

Our research team previously found that a few participants had unmet expectations of the study's financial contributions for time spent participating in the trial and for transportation costs, which in some cases led to their withdrawal from the study [25]. However, it is difficult for the sponsor and investigators to implement the transport contribution for unscheduled events, a recurrent suggestion arising from this evaluation study, especially when there are no major safety concerns. The transportation costs, along with cash payments required for health services, generally deter many households from accessing (emergency) care, especially in LMICs[6]. Other researchers have pointed out that from volunteers' perspectives, the true incidence rate of AE during trial participation exceeds the frequencies reported by investigators, and that underreporting is common among healthy research volunteers [8]. This might lead to biased reporting, where more severe AE are reported while less severe ones may go unreported, potentially skewing the distribution of AE severity. Both our qualitative and quantitative findings reveal that the free AC services provided during research participation contributed to participants actually accessing the needed care, be it at the trial site or closer to their residence location, resulting in a more realistic AE reporting during trial participation [13].

Although the participant population consisted of healthcare workers and frontliners who were either formally trained in, or at least occupationally familiar with, conventional medicine practices, the financial impact of AE makes the presence and recourse to traditional medicine practitioners in this area frequent [12]. They are often more accessible, offer the most – or only – affordable treatment option and tend to accept different payment forms, which is particularly attractive and convenient for low-income households [5]. Additionally, seen their professional background, our participants might more easily self-medicate when experiencing AE, especially when this is facilitated by easily available over-the-counter medication [29]. Although traditional medicine practices and self-medication are known to impact on AE health seeking behavior [13], when enquiring about their preferred treatment location, the majority of our telephone survey respondents (93.5%, n = 272) still referred to the GRH in Boende.

Various aspects of the trial environment have shown to influence the well-being of trial participants [8]. However, in the context of Boende, it became evident that the concept of 'well-being', or the French term 'bien-être', does not translate easily into Lingala, with several possible translations, each carrying different nuances. As a result, participants in this cultural and geographic setting were generally unfamiliar with the concept as it is understood by the clinical researchers. This issue was highlighted by the interpreter, who noted difficulties when translating interview questions into Lingala, and was further reflected in the survey results, where a considerable number of participants (N=41) selected 'no opinion' when asked how the AC support contributed to their overall well-being during the trial. Unfortunately, due to time constraints, these questions could not be adequately piloted before the interviews began; this will be taken into account when planning future studies in this context.

The acceptability study provided valuable insights into participants' views and perceptions on the study's AC support. Notably, there were instances where participants' expectations did not fully align with the intended scope of the policy. For example, some participants expressed disappointment that AC support was not continued after the trial, suggesting a misunderstanding of the policy's purpose and duration. This highlights the importance of establishing and reinforcing realistic expectations from the outset and throughout the study. The fact that such misconceptions were expressed by healthcare-affiliated participants underscores the need for more robust communication strategies during the informed consent process and ongoing engagement, to ensure that the objectives and limitations of the study and its AC provisions are clearly understood. However, the observed perception of research participation as a strategy to get better access to quality healthcare cannot be fully eliminated, as long as quality and affordable health care is not locally available outside the study [27].

Our findings highlight the significance of comprehensive participant care within the context of clinical research. As a result, five actionable recommendations were listed (Table 2). The development and communication of formalized AC

**Table 2. Key strategic recommendations for Ancillary Care (AC) policy development and adaptation for clinical trials conducted in resource-constrained settings.**

| 1 | Implement formalized AC policies in clinical studies | To enhance participant enrolment and follow-up, positively support health-seeking behaviour, ensure accurate adverse event reporting, and fulfil the ethical responsibilities of both the sponsor and the research team, clinical trials – especially in low- and middle-income countries – should establish formalized AC policies. |
|---|---|---|
| 2 | Introduce AC policies at the start of the study | Well-informed AC policies should be agreed and implemented from the start of the study. Delaying the introduction of these policies can lead to perceptions of unfairness and reduce participants' appreciation, particularly among those who did not initially receive this support. |
| 3 | Communicate AC policies clearly and promptly | To enhance communities' and participants' understanding and manage expectations, AC policies (including reimbursement procedures) should be clearly explained at the outset and throughout the study. This transparent communication is crucial for securing participant trust and endorsement. |
| 4 | Foster collaboration among stakeholders in resource-constrained settings | The complexity of conducting research in resource-limited environments requires close collaboration among donors, sponsors, researchers, community representatives, local health actors, and ethics committees. This is vital for co-designing adequate and feasible AC policies that consider the responsibilities of all stakeholders and prioritize the well-being of trial participants and engagement with their communities. |
| 5 | Use acceptability studies to guide AC policy revisions | The findings from acceptability studies should be used to inform and refine AC policies for future research. This ensures that the concerns and needs of the community are adequately addressed, leading to more effective and ethically sound research practices. |

policies is crucial, as it will enhance participant engagement, ensure fairness, and uphold the ethical responsibilities of research sponsors and investigators. Introducing these policies early, along with effective collaboration among stakeholders and upfront engagement with the community, is essential for addressing the complexities of resource-constrained settings. Moreover, the continuous revision of AC policies based on study outcomes is necessary to meet community needs and improve future research.

This study faced several limitations. First, we were unable to link participant surveys to specific case report forms for reported and treated (S)AE, limiting our ability to make broad conclusions about participants' overall satisfaction with the different types of support (e.g., reimbursements, direct payments, provision of medication) and to differentiate between AC support for AE and SAE. Second, the involvement of local trial staff (RAM and BLB) and a sponsor team member (GL) in the qualitative research may have introduced bias. Their roles as interpreters during interviews might have influenced participants' responses due to desirability bias, both in the way answers were given and translated. Additionally, the fact that study staff recorded survey answers could have further impacted the responses. Third, the translation of interview and survey questions across languages, as well as the back-translation of participants' responses, may have led to the loss of certain meanings or sensitivities. Finally, the absence of an external evaluation of the AC policy is another limitation, as the (co-)developer of the AC policy also led the evaluation study, conducted interviews, and analyzed the data.

## Conclusion

This study assessed participant perceptions, attitudes, and acceptability of the AC policy within the Ebola vaccine trial in Boende. Overall, the policy was highly endorsed and underlines the importance of providing AC during clinical trials in resource-constrained settings. Participants reported multifaceted benefits, although limitations were equally identified. Regularly revising AC policies based on study outcomes and findings from acceptability studies is essential to prioritize the well-being of trial participants and to address the expectations and concerns of the community. We hope these findings and recommendations offer valuable insights for future clinical research and AC policy development, specifically within the context of resource-constrained settings.

## Supporting information

**S1 Fig. Survey ratings (questions 7–10) from participants who experienced adverse events, assessing perceived consequences of the absence of ancillary care (AC) support.**
(TIF)

## Acknowledgments

We gratefully acknowledge the hard work of the local trial staff for the data collection, especially Dr. Emmanuel Esanga, Dr. Michael Bojabwa Mondjo, and Dr. Jimmy Mpato Manga. A special thanks goes to Rebecca Asieli Malaza and Benedicte Liuba Bala for assisting with the participant interviews by translating questions into Lingala, to Dr. Nana Yongo, who conducted the transcriptions and translations (Lingala to French) of these audio records, as well as Dr. Antea Paviotti who provided quality control of survey questions and assistance with (English to French) translations.

## Author contributions

**Conceptualization:** Gwen Lemey.

**Data curation:** Solange Milolo.

**Formal analysis:** Ynke Larivière.

**Methodology:** Gwen Lemey.

**Project administration:** Vivi Maketa.

**Supervision:** Sibyl Anthierens.

**Validation:** Raffaella Ravinetto, Vivi Maketa, Sibyl Anthierens.

**Writing – original draft:** Gwen Lemey.

**Writing – review & editing:** Ynke Larivière, Trésor Zola Matuvanga, Maha Salloum, Patrick Mitashi, Pierre Van Damme, Raffaella Ravinetto, Jean-Pierre Van geertruyden, Hypolite Muhindo-Mavoko, Vivi Maketa.

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
