## [Decision Letter · Decision Letter 0]

PONE-D-24-42834“We have to amplify what we saw at EBOVAC” – A mixed methods approach to assess participant acceptability of ancillary care in an Ebola vaccine trial in the Democratic Republic of the CongoPLOS ONE

Dear Dr. Lemey,

Thank you for submitting your manuscript to PLOS ONE. After careful consideration, we feel that it has merit but does not fully meet PLOS ONE’s publication criteria as it currently stands. Therefore, we invite you to submit a revised version of the manuscript that addresses the points raised during the review process.

We look forward to receiving your revised manuscript.

Kind regards,

Omar Enzo Santangelo

Academic Editor

PLOS ONE

Journal Requirements:

4. In this instance it seems there may be acceptable restrictions in place that prevent the public sharing of your minimal data. However, in line with our goal of ensuring long-term data availability to all interested researchers, PLOS’ Data Policy states that authors cannot be the sole named individuals responsible for ensuring data access (http://journals.plos.org/plosone/s/data-availability#loc-acceptable-data-sharing-methods).

Additional Editor Comments:

Dear Authors, the manuscript needs minor revisions, please respond point by point to the reviewers' requests.

Reviewers' comments:

Reviewer's Responses to Questions

**Comments to the Author**

1. Is the manuscript technically sound, and do the data support the conclusions?

Reviewer #1: Yes

Reviewer #2: Yes

2. Has the statistical analysis been performed appropriately and rigorously? 

Reviewer #1: Yes

Reviewer #2: Yes

3. Have the authors made all data underlying the findings in their manuscript fully available?

Reviewer #1: Yes

Reviewer #2: No

4. Is the manuscript presented in an intelligible fashion and written in standard English?

Reviewer #1: Yes

Reviewer #2: Yes

5. Review Comments to the Author

Reviewer #1: This paper describes the participant experiences in an Ebola vaccine trial. Health care availability is poor in the region and the study determined that an ancillary care program would be helpful for participants. however, this was not implemented until later in the study.

1. The team should do more to clarify the timeline of initial vaccinations and boosters to the availability of the AC program. With the program starting in the final year of the program - it seems to be after the time when most would experience treatment related AEs. It is not clear if the AEs and SAEs being evaluated are study related in general, or if they are health problems experienced by participants (for instance, there is apparently a leg injury in the study and that seems unlikely to be trial related). It is not necessarily important to know which are related or not. But surely, care for related AEs would be considered the highest priority for the study.

2. The timeline in figure 1 is not clear. The big bubbles are colorful, but are not lined up with the time period during which they were administered. When were study events, first vaccines, booster vaccines? Are they all completed by the time the AC was implemented? The colored bars are not as obviously associated with the different bubbles and I only noticed they were numbered after staring at the figure for some time. The bubbles are also not relative in size to the sample size and thus a bit misleading. 17 interviews should have a much smaller bubble.

3. The bar graphs of the survey responses are not all necessary. The telephone survey results do not need a bar graph - everyone was happy. The Q1-6 is nice, the Q7-8 and Q10 are also not very informative. Q9 does have more diversity of answers, but it is equally well represented in the text.

4. a. Table 1 is confusing in parts. There are 700 participants in the vaccine study, with 370 individuals reporting an AE? Or 370 reported AEs?

b. 185 of those surveyed - our of 290? - had an AE.

c. What are the demographics of the surveyed participants who did not have an AE? Two columns would be good for that category as is done for the telephone interviews.

d. Then the telephone survey participants are divided by with and without an AE, but one group is without self-reported AE, and with self reported unsupported AE. Were the ones without a self-reported AE eligible for care, if they had needed it?

e. The racial distribution seems unnecessary as the study was all Black Africans.

f. The differences in AEs by sex are striking. Given the differences in self-report bias that might occur in this study it seems appropriate to discuss these differences. Men made up 3/4 of those with AEs. How does this relate to the overall study sample?

g. What does "medical history" refer to in the table? It is different by groups.

h. what are arms 1 and 2? Did the Arm 2 participants get any injections while the study was underway?

5. The study is a very nice evaluation of perceptions. I imagine there is heavy bias towards socially desirable reporting. This was mentioned in the discussion, but ways to mitigate this should be considered for future studies. Anonymous reporting might be less biased. The telephone interviews show a strong leaning towards that. But in an area with little health care, the AC program was probably very helpful. One element the study team should discuss a bit more is the disconnect between the perceptions of what the AC program was for - and what the participants wanted or expected. Establishing reasonable expectations is key when enrolling participants. The concept that the AC program was not "long lasting enough", that they would be gone tomorrow, that is a key misconception. And these participants were associated with the health care setting - many being CHWs and professionals. If these participants were misinformed, it is vital that programs do an even better job instructing their participants as to what they might expect.

6. There are a lot of quotes from participants, some may even be identifying. How many people had a serious leg injury in the study? It seems that the study team might want to obscure that a bit more. The editors may wish to reduce the number of quotes - but they are interesting and informative. Usually one doesn't get to see the primary results. I enjoyed reading them.

The paper is well written, the writing style is clear and easy to read. I anticipate that the study team can make the tables and figures more crisp and address some of the confusion around the parent study timeline and the AC evaluation timeline.

Reviewer #2: The authours have addressed an important topic and sused appropriate methods.However the follwing areas should be addresed/clarified:

1.The tittle should be revised to capture what is written in the manuscript.Thoughout the manuscript,in addition to acceptability,the words perception,attitudes and even safisfaction come up.These should be included in the title especially attitudes and percecption,the opening statement of the discussion section,makes a good case for their nclusion in the title. In addition the word``policy ``should be added to ancillary care in the tittle.

2.In the abstract,the indroduction should be beefed up,they authours cang et a brief statement from the first paragraph of the main intoduction. The study objective should also be stated in the abstract.

3.The introduction was well staed but the objectives which have been stated in lines 85-88and that in lines91-92 should be harmined and sorted out which ones fit well with the study and study tittle.and the objectives should stated at the end of the introduction.The statement ``This paper............among trialparticipnts`` can beleft out

4.Study design,sampling and data collection: The methods from lines 101-159 need to be revised and state clearly how the interviews were conducted and how the participants for diferent intreviews were selected andhow many were allocated for each surveyand whic questions wre asked, on the contrary,fromthe results,it indicates the partipants were responding to specific questions.In the methods the numbers of study participants shoud be clearly statedi because it comes a challenge when it comes to the results section.The diffent cadres of health workersare not stated in themothods but they appear in the resuktsintable 1.who are`frontliners?``

5.Results:At the begining of te results section the numbersof the total study participnts and the different categories are not stated. Howver in table 1 on the social demographic and baeline charactersoistics, these numbers comeup.The othaurs should clearly clarify n these bnumbsers fr exampleinthe astract it says the total number of participants whohad AEs were 185 !! As stated in the methods section these numbers should be clarified.

Otherwise the results werefairly wellp resented.however in some instances,the results are mixed with methods,the me thods shouldnt be includ ed in the results e.g lines 217-225,methods are being mentined together with results,same case with lines 239-244. Also in the qualitative results,there is need to teese out the results from what was presented.

e.g.lines323-326and 340-343. Is possible to further summerise the this section?

6.Discussion:Theauthur discussed teir results well .However is the state inline 378-380 and refering totable results from their study or recomanation,let it be put in appropriate section.it is unsuual to have a table in thediscusionsection.

7.Conclusions&recommedations:Thesshould be revised to be in line with te study ttile and objectives and study resukts respectively.

6. PLOS authors have the option to publish the peer review history of their article (what does this mean?). If published, this will include your full peer review and any attached files.

Reviewer #1: No

Reviewer #2: **Yes: **MworoziEdison Arwanire

---

## [Author Response · Author response to Decision Letter 1]

5 May 2025

Reviewer #1: This paper describes the participant experiences in an Ebola vaccine trial. Health care availability is poor in the region and the study determined that an ancillary care program would be helpful for participants. however, this was not implemented until later in the study.

1. The team should do more to clarify the timeline of initial vaccinations and boosters to the availability of the AC program. With the program starting in the final year of the program - it seems to be after the time when most would experience treatment related AEs. It is not clear if the AEs and SAEs being evaluated are study related in general, or if they are health problems experienced by participants (for instance, there is apparently a leg injury in the study and that seems unlikely to be trial related). It is not necessarily important to know which are related or not. But surely, care for related AEs would be considered the highest priority for the study.

R/ thank you for your suggestion. We have adapted Figure 1 to include details on the timeline of vaccinations, active and passive phases, as well as the timepoint of the AC policy implementation and acceptability study. Also, the adapted figure will clarify that both medical events related to the study/investigational product (booster vaccination was still part of trial arm 2 participants’ trajectory) and the unrelated medical events experienced by participants during the last year of the trial were treated and/or supported. For clarity, the scope (and limitation) of the AC policy was better described in lines 84-88 (manuscript version with track changes), in line with lines 80-81, where we described the policy’s goal to “address, treat and support trial participants experiencing medical events, irrespective of a causal link with the research intervention”. The wording ‘whether study-related or not’ is also added in the abstract.

2. The timeline in figure 1 is not clear. The big bubbles are colorful, but are not lined up with the time period during which they were administered. When were study events, first vaccines, booster vaccines? Are they all completed by the time the AC was implemented? The colored bars are not as obviously associated with the different bubbles and I only noticed they were numbered after staring at the figure for some time. The bubbles are also not relative in size to the sample size and thus a bit misleading. 17 interviews should have a much smaller bubble.

R/ Thank you for your input; we hope that the adaptations made to Figure 1 address these remarks.

3. The bar graphs of the survey responses are not all necessary. The telephone survey results do not need a bar graph - everyone was happy. The Q1-6 is nice, the Q7-8 and Q10 are also not very informative. Q9 does have more diversity of answers, but it is equally well represented in the text.

R/ we understand your point and suggest to:

1) remove Figure 2 referring to the telephone survey responses, as the results were also outlined in the text under Results – Survey results.

2) Keep bar graph with Q1-6 (Figure 3, renamed to ‘Figure 2’);

3) Move bar graph Q7-10 to supplementary material for the reader’s reference, as the main results are described in the text.

References to these figures (and their numbering) were adapted in the main text, as well as a clarification in line 256-257, and a correction of a small percentage error (line 263). We hope these changes are satisfactory.

4.

a. Table 1 is confusing in parts. There are 700 participants in the vaccine study, with 370 individuals reporting an AE? Or 370 reported AEs?

R/ Thank you for pointing out the potential confusion in Table 1 regarding the number of participants and adverse events (AEs). We appreciate the opportunity to clarify this point. As noted, the Ebola vaccine trial started its activities with 698 vaccinated participants (clarified in line 60). By the third year—when the Ancillary Care policy was implemented and the acceptability study began—655 participants remained enrolled. This information was not previously included in the main text and has now been added for clarity (lines 122–123, and in the title of Table1, line 233). With regard to the AEs, a total of 370 individual participants reported experiencing at least one AE. To eliminate ambiguity in Table 1, we have updated the wording to explicitly indicate that 370 individuals reported AEs, rather than implying 370 total AEs (some individuals may have reported more than one AE). Additionally, we have added the symbol “≥” in the relevant part of the table to emphasize that the count refers to participants who reported one or more AEs. We hope these revisions clarify the information and improve the readability of the table and associated text.

b. 185 of those surveyed - our of 290? - had an AE.

R/To clarify: a total of 290 surveys were collected from 185 unique individuals who had experienced at least one AE (cf. line 244-245) in the final operational year of the trial. This means that some participants may have experienced (and reported) multiple AE.

c. What are the demographics of the surveyed participants who did not have an AE? Two columns would be good for that category as is done for the telephone interviews.

R/ A column was added (first column in the Table 1) with demographics of participants without reported AE.

d. Then the telephone survey participants are divided by with and without an AE, but one group is without self-reported AE, and with self reported unsupported AE. Were the ones without a self-reported AE eligible for care, if they had needed it?

R/ During the follow-up telephone calls, the participants were specifically asked whether they had experienced a medical event that was not supported by the AC policy. The ones that were reported, and (potentially) received AC treatment or support, would fall under the category “Participants with ≥ 1 reported AE (N=370)”.

e. The racial distribution seems unnecessary as the study was all Black Africans.

R/indeed, this distribution is now removed.

f. The differences in AEs by sex are striking. Given the differences in self-report bias that might occur in this study it seems appropriate to discuss these differences. Men made up 3/4 of those with AEs. How does this relate to the overall study sample?

R/ Thank you for this important observation regarding the difference in AE reporting by sex. We agree that these differences merit discussion. For your reference, we added the following note in the Table: “The higher absolute number of AE reporting in males reflects the higher proportion of males in the Ebola vaccine trial at the time of the acceptability study (n=508; 77.6% male versus n=147; 22.4% female) (13).”

Additionally, as part of the broader quantitative evaluation of the AC policy (Lemey G, Larivière Y, et al. An ancillary care policy in a vaccine trial conducted in a resource-constrained setting: evaluation and policy recommendations. doi: 10.1136/bmjgh-2024-015259); we found that while age and medical history at study entry did not significantly influence AE reporting, men were approximately one third less likely than women to report health problems. This suggests that self-report bias, including gender-related reporting behaviours, may partially explain the observed differences. Interestingly, although men made up a majority of AE reporters in this subset, this does not directly reflect higher incidence, but rather possible variations in reporting behavior tied to visit frequency or occupational context. For example, participants working in health facilities were also less likely to report AEs than those in community roles. Additionally, participants in Arm 2, who had more scheduled visits, were more than twice as likely to report health problems as those in Arm 1, likely due to increased opportunities for disclosure. We have added a concise summary of this context to the manuscript to clarify this point (lines 227–230).

g. What does "medical history" refer to in the table? It is different by groups.

R/ this refers to the relevant medical history (considering trial participation) that was registered at the time of main trial enrollment. This did not impact AE reporting, and was made clear for the reader in lines 227-230.

h. what are arms 1 and 2? Did the Arm 2 participants get any injections while the study was underway?

R/ In the main trial, the vaccine booster dose was administered one (Arm 1) or two (Arm 2) years after the first dose. We hope this remark has been clarified by adapting Figure 1 (also referring to your first remark), outlining the timepoints of the vaccinations/boosters per trial arm and the AC policy evaluation study. A reference to this figure was added in lines 187-189: “However, only participants from trial arm 2 could be included (target sample size = 331), as the arm 1 participants already had their last safety phone call at a different timepoint (Fig 1).”

5. The study is a very nice evaluation of perceptions. I imagine there is heavy bias towards socially desirable reporting. This was mentioned in the discussion, but ways to mitigate this should be considered for future studies. Anonymous reporting might be less biased. The telephone interviews show a strong leaning towards that. But in an area with little health care, the AC program was probably very helpful. One element the study team should discuss a bit more is the disconnect between the perceptions of what the AC program was for - and what the participants wanted or expected. Establishing reasonable expectations is key when enrolling participants. The concept that the AC program was not "long lasting enough", that they would be gone tomorrow, that is a key misconception. And these participants were associated with the health care setting - many being CHWs and professionals. If these participants were misinformed, it is vital that programs do an even better job instructing their participants as to what they might expect.

R/ We appreciated this insightful comment and reflection on the potential for socially desirable reporting, as well as the important point about the disconnect between participants expectations and the intended scope of the AC programme. We have taken note of the reviewer’s suggestion regarding anonymous reporting as a potential strategy to reduce socially desirable responses and will consider this approach in the design of future studies.

We also agree that despite participants’ affiliation with health care settings, such as being HCWs, there were notable misconceptions about the longevity and purpose of the AC policy. As suggested, this highlights a critical need for clearer communication and expectation setting throughout the study, especially during the informed consent process and subsequent participant engagement. However, these misconceptions probably also underscore the intrinsic challenges of informed consent in socio-economically vulnerable populations who can see research participation as a strategy to access a better level of care, as observed by some of the co-authors in the past in an urban setting in DRC (ref Kalabuanga M, Ravinetto R, Maketa V, et al. The challenges of research informed consent in socio-economically vulnerable populations: a viewpoint from the Democratic Republic of Congo. Dev World Bioeth 2015; 16(2): 64-69).

We have expanded the discussion accordingly in lines 621-631 with the following addition: “The acceptability study provided valuable insights into participants’ views and perceptions on the study’s AC support. Notably, there were instances where participants’ expectations did not fully align with the intended scope of the policy. For example, some participants expressed disappointment that AC support was not continued after the trial, suggesting a misunderstanding of the policy’s purpose and duration. This highlights the importance of establishing and reinforcing realistic expectations from the outset and throughout the study. The fact that such misconceptions were expressed by healthcare-affiliated participants underscores the need for more robust communication strategies during the informed consent process and ongoing engagement, to ensure that the objectives and limitations of the study and its AC provisions are clearly understood. However, the observed perception of research participation as a strategy to get better access to quality healthcare cannot be fully eliminated, as long as quality and affordable health care is not locally available outside the study(25)”.

6. There are a lot of quotes from participants, some may even be identifying. How many people had a serious leg injury in the study? It seems that the study team might want to obscure that a bit more. The editors may wish to reduce the number of quotes - but they are interesting and informative. Usually one doesn't get to see the primary results. I enjoyed reading them.

The paper is well written, the writing style is clear and easy to read. I anticipate that the study team can make the tables and figures more crisp and address some of the confusion around the parent study timeline and the AC evaluation timeline.

R/ We have modified the quote from the participant with the leg injury (lines 336-340), obscured the dates from all the quotes and removed 4 quotes throughout the results section that expressed similar ideas or overlapping content. We are grateful for your constructive feedback and valuable review. We appreciated your suggestions and hope to the revisions adequately address your concerns !

* * *

Reviewer #2: The authours have addressed an important topic and sused appropriate methods.However the follwing areas should be addresed/clarified:

1.The tittle should be revised to capture what is written in the manuscript.Thoughout the manuscript,in addition to acceptability,the words perception,attitudes and even safisfaction come up.These should be included in the title especially attitudes and percecption,the opening statement of the discussion section,makes a good case for their nclusion in the title. In addition the word``policy ``should be added to ancillary care in the tittle.

R/ thank you for this suggestion, we have revised the title to better capture the scope of the study “We have to amplify what we saw at EBOVAC” – Assessing Participant Perceptions, Attitudes, and Acceptability of an Ancillary Care Policy in an Ebola Vaccine Trial in the Democratic Republic of the Congo: A Mixed Methods Study.

2.In the abstract,the indroduction should be beefed up,they authours cang et a brief statement from the first paragraph of the main intoduction. The study objective should also be stated in the abstract.

R/ thank you for this suggestion, this information is now also added to the abstract, lines 26-32 (manuscript version with track changes): “In a vaccine trial conducted between 2019 and 2022 in Boende, a remote, resource-constrained area of the Democratic Republic of the Congo, our research team developed an ancillary care (AC) policy to provide adequate care and follow-up for concomitant adverse events (AE), whether study related or not. The trial aimed to assess the safety and immunogenicity of an Ebola vaccine regimen among approximately 700 healthcare providers and frontliners to strengthen outbreak preparedness in this Ebola-endemic region, where access to healthcare is severely limited by poverty, weak infrastructure, and an overstretched health system.”

3.The introduction was well staed but the objectives which have been stated in lines 85-88and that in lines91-92 should be harmined and sorted out which ones fit well with the study and study tittle.and the objectives should stated at the end of the introduction.The statement ``This paper............among trialparticipnts`` can beleft out

R/ Thank you for this helpful observation regarding the clarity and placement of the study objectives. In response, we have carefully reviewed the objectives stated in lines 93–102, and have harmonized them to ensure consistency with the study’s scope and revised title. The objectives have now been clearly and succinctly stated at the end of the introduction, as per your suggestion. Additionally, the sentence beginning with “This paper…” has been removed, as recommended, to improve the overall flow and focus

---

## [Decision Letter · Decision Letter 1]

“We have to amplify what we saw at EBOVAC” – Assessing Participant Perceptions, Attitudes, and Acceptability of an Ancillary Care Policy in an Ebola Vaccine Trial in the Democratic Republic of the Congo: A Mixed Methods Study

PONE-D-24-42834R1

Dear Dr. Lemey,

We’re pleased to inform you that your manuscript has been judged scientifically suitable for publication and will be formally accepted for publication once it meets all outstanding technical requirements.

Kind regards,

Omar Enzo Santangelo

Academic Editor

PLOS ONE

Additional Editor Comments (optional):

Reviewers' comments:

Reviewer's Responses to Questions

**Comments to the Author**

1. If the authors have adequately addressed your comments raised in a previous round of review and you feel that this manuscript is now acceptable for publication, you may indicate that here to bypass the “Comments to the Author” section, enter your conflict of interest statement in the “Confidential to Editor” section, and submit your "Accept" recommendation.

Reviewer #2: All comments have been addressed

2. Is the manuscript technically sound, and do the data support the conclusions?

Reviewer #2: Yes

3. Has the statistical analysis been performed appropriately and rigorously? 

Reviewer #2: Yes

4. Have the authors made all data underlying the findings in their manuscript fully available?

Reviewer #2: No

5. Is the manuscript presented in an intelligible fashion and written in standard English?

Reviewer #2: Yes

6. Review Comments to the Author

Reviewer #2: The authours have appropriately addressed and responded to all the comments/recommendations by the reviewers.

7. PLOS authors have the option to publish the peer review history of their article (what does this mean?). If published, this will include your full peer review and any attached files.

Reviewer #2: **Yes: **Edison Mworozi Arwanire

---

## [Editor Report · Acceptance letter]

PONE-D-24-42834R1

PLOS ONE

Dear Dr. Lemey,

I'm pleased to inform you that your manuscript has been deemed suitable for publication in PLOS ONE. Congratulations! Your manuscript is now being handed over to our production team.

Kind regards,

on behalf of

Dr. Omar Enzo Santangelo

Academic Editor

PLOS ONE